# Geometric Parameter Identification of Large Bent Pipes Using a Single-View Vision System

**DOI:** 10.3390/s25175420

**Published:** 2025-09-02

**Authors:** Krzysztof Borkowski, Dariusz Janecki, Jarosław Zwierzchowski, Dawid Sebastian Pietrala

**Affiliations:** 1Department of Automation and Robotics, Kielce University of Technology, al. Tysiąclecia P. P. 7, 25-314 Kielce, Poland; djanecki@tu.kielce.pl (D.J.); dpietrala@tu.kielce.pl (D.S.P.); 2Institute of Electronics, Lodz University of Technology, al. Politechniki 10, 90-590 Łódz, Poland; jaroslaw.zwierzchowski@p.lodz.pl

**Keywords:** optical measurement, 2D vision in 3D analysis, geometric parameter identification, industrial metrology

## Abstract

This paper describes methods of determining important measurement parameters of large bent pipes with diameters of up to 1.2 m for heavy industry, which can be obtained instantly from a vision system. The article presents, in detail, modeling methods of the bending angle, radius, and straight sections of the bent pipe. The system is able to detect the start and end of such sections, which is novel in automatic pipe measurement. The article also demonstrates the use of a modified Hough transform in line and curve fitting and the necessary image preprocessing. The complete system operates on distortion models and image projection dedicated for pipe models with images taken from a single camera.

## 1. Introduction

Vision systems play a crucial role in modern manufacturing, enabling continuous monitoring and real-time adjustment of production parameters to ensure compliance with increasingly stringent quality standards. Despite significant technological progress, there remains a noticeable lack of simple and effective measurement systems dedicated to applications in heavy industry.

The purpose of this study is to develop algorithms for a vision-based measurement system intended to inspect the dimensions and shapes of large-scale bent pipes used in heavy-industry applications. A key objective is also to evaluate the measurement accuracy of the proposed system using data acquired from a single-camera setup and to compare the results with those obtained from a coordinate measuring machine (CMM).

The use of low-cost 2D vision systems for the dimensional inspection of 3D objects is a worthwhile approach, offering a balance between system simplicity, cost-effectiveness, and acceptable measurement accuracy in industrial environments.

Our system is designed to verify the dimensional conformity of nominal cylindrical and toroidal geometries from a single image and is not intended to detect defects such as ovalization or local variations in the radius of the bend.

The authors previously published an ICCS 2023 paper describing the method of mathematical modeling of bent pipes and camera calibration, also presenting quantitative and qualitative measurements [1]. The focus was on single- and double-bending models in one plane, and the perspective projection for images obtained from the vision system was described there. In this work, we extend that research and present methods for estimating parameters relevant to heavy-industry inspection. It was assumed that the bent pipe model is already known and that an appropriate perspective projection was made.

### 1.1. List of Main Contributions

The new elements that are worth emphasizing in this paper are the following: (1) algorithms for the necessary image pre-processing, (2) an algorithm for fitting straight lines or curves to the contour of the bent pipe and a modification of Hough transform, and (3) a mathematical method for determining the pipe diameter and bending angle, as well as the beginning and end of straight sections in bent pipes.

### 1.2. Related Works

A wide range of methods for shape measurement and classification have been developed, which can be broadly divided into contact and non-contact techniques [2,3]. Contact methods, such as coordinate measuring arms [4,5], offer high accuracy, but they require manual operation and are not suitable for automated inspection of large-scale bent pipes. As a result, non-contact optical methods—both active (e.g., laser scanning, structured light projection [6,7]) and passive (e.g., photogrammetry, stereovision [8,9])—have become increasingly popular for the measurement and inspection of complex pipe geometries. However, the application of active optical methods, such as fringe projection [10] to long metallic pipes, is limited by issues with specular reflections and the difficulty of ensuring uniform projection and calibration over large areas [11]. Consequently, recent research has focused on vision-based and multi-view approaches, as summarized below.

In recent years, several studies have been published on optical inspection and measurement systems for bent pipes and tubes. Most of these systems reconstruct the 3D surface of the pipe using passive multi-view approaches.

In [12], a preliminary discretization of the point cloud in circles was used. In experiments carried out in a measurement space of 1850×1200×500mm, a standard deviation of 0.12mm was achieved for dimensional length A (Figure 1 at Ref. [12]), which measured 83.94mm. The relative uncertainty for the dimension length was 0.14%.

The study presented in [13] introduced a photometric linearization method to eliminate light reflections on the surface of the pipe and a method to determine the length of the centerline. In the experiments, an eight-camera vision system with a measurement area of 850×850mm was utilized. During the measurement, a standard deviation of 0.167mm was achieved for the length of the centerline, which was measured as 373mm. The relative uncertainty for measuring the length of the centerline was 0.04%.

The system described in [14] consists of 16 cameras with a measurement volume of 1850×1200×500mm. The experiments carried out yielded an average squared measurement error of 0.5mm for a bending radius of 60mm. The relative uncertainty for the radius was 0.83%.

The work presented in [15] describes a pipe measurement system based on stereo vision, incorporating a perspective model of the pipe to enhance the accuracy of the reconstructed spatial axis.

Huang [16] conducted an error analysis of 3D reconstruction for bent pipes using a multicamera setup divided into stereo pairs. In addition, a backlit work table was used to improve the visibility of the pipes features.

In [17], a system was presented to reconstruct the shape of pipes from images captured by a three-view vision system. The study accounted for perspective distortions for each camera, and the pipe surfaces were described using circles of the same diameter. An average squared measurement error of 0.125mm was obtained for pipes with a bending radius of 4.16mm. The relative uncertainty of the measurement of the radius of the bend was 3.00%.

In [18], a multi-camera system with a working area of 600×1200mm, comprising eight cameras, was introduced. For the key points determined on the pipe, an average standard deviation of 0.025mm was obtained with a pipe length of 819.922mm.

Active methods have also been explored, particularly those based on laser scanning. In [19], a 3D model of a pipeline was generated from a point cloud, achieving a mean standard deviation of 3.40mm.

The study in [20] presents a system for continuous monitoring of the bending angle both during and immediately after the forming process. A laser-based device mounted at the end of the pipe was used for this purpose.

Classical single-view metrology establishes how absolute dimensions can be inferred from a single perspective image given minimal geometric references and calibration constraints [21]. Robust ellipse or circle modeling in perspective is widely used for cylindrical objects and underpins many industrial vision solutions [22]. More recent work explores the combination of geometric models with learning-based ellipse prediction for improved stability [23]. In contrast, deep learning approaches in pipe inspection focus primarily on detecting and segmenting defects in CCTV or radiographic imagery [24,25] and typically do not report the absolute geometric parameters of large bent pipes from a single image. Our model-based, single-camera pipeline complements these efforts by providing explicit parameter estimation (diameter, bend angle and radius, straight section lengths) with accuracy referenced to a CMM.

## 2. Materials and Methods

The calculations and algorithms described in this work were implemented using Wolfram Mathematica (Version10.0); this is a research prototype, not a finalized engineering implementation.

Image acquisition was carried out with a camera PIXELINK PL–D7715CU (Pixelink, Inc., Ottawa, ON, Canada), featuring a 15Mpix CMOS sensor (4608×3228px resolution), equipped with a TECHSPEC 58–000 fixed focal length lens with a focal length of 8.5mm.

For the purposes of this study, a laboratory measurement station was prepared on a 1:4 scale, replicating a real marking table used on the production line. The table was made of laser-cut steel sheets to reproduce real working conditions, including the presence of gaps between the table segments and surface imperfections. The dimensions of the laboratory table are 1000×1745mm, and the distance from the camera to the surface of the table is 2300mm. Figure 1 shows a side view of the measurement station.

We measure the parameters of a bent pipe lying freely on the measuring table based on its external contour observed on the camera image. Consider a vision system that uses two example images taken at the measuring station. The images have a resolution of 4608×3228px and are stored in the RGB color space. Our goal is to determine the parameters of the bent pipe. We assume that the measured object is a combination of cylindrical and toroidal surfaces with a constant radius of the cross-section. This makes it possible to measure the parameters of the object based on the image obtained from one camera setting. It should be mentioned here that this is an approximation; in reality, the arc of the bent pipe consists of many smaller arcs with different radii, and in addition, the pipe undergoes ovalization (change of the shape of the pipe cross section from circular to elliptical).

The parameters subjected to control are marked in Figure 2 and are as follows: bending angle *γ*, bending radius *R*, straight sections lengths $l_{1}$, and pipe diameter *D*.

The desired parameter values can be determined based on the knowledge of the edge points of the pipe determined in the plane containing its centerline (Figure 2). In the case where the pipe with one bent arc lies freely on the measuring table, we obtain the edge points by projecting it straight onto the plane of the measuring table. The edge points—the contour of the pipe—are also visible in the digital photo from the station. However, it should be emphasized that the outline of the pipe observed from the camera point can be interpreted as its central (perspective) projection onto the surface of the table.

### 2.1. Processing and Segmentation of Images

The segmentation process leads to the determination of identifiable sets of points from a digital image, which in this case are sets of edge points of bent pipes lying freely on the measurement table.

Consider a vision system that utilizes two sample images captured at the measurement station. Both images have a resolution of 4608×3228px and are stored in the RGB color space.

The first step is to remove the background of the image with the pipe by subtracting the preliminary image on which the object under study is not located.

To improve contrast and reduce differences between images, we perform histogram equalization for both images [26]. This method flattens the image histogram while enhancing the details. The transformation is achieved by linearizing the cumulative distribution function of pixel brightness probability in the image and is calculated separately for each RGB color component. The resulting image is shown in Figure 3.

In the next step, we preliminarily identify areas with pipes by calculating the difference between the images. In the resulting image, pipes along with fragments of the measurement table, which are false detections, are visible.

To correct for errors in objects, we apply the morphological closing operation (for the color image [27]). We obtained a color image with distinguishable regions where the bent pipes are located.

Next, we convert the image to grayscale and perform its binarization, with the threshold determined based on the histogram.

The binary image obtained (Figure 4) consists of uniform regions (sets of points), each associated with a single pipe. We can assume that the edges of the pipes are located near the contour of the binary mask; however, it should be emphasized that they may also include the shadow of the pipe. Therefore, we will use them to select the object and reject edges that are not associated with the pipe.

The edge points of the pipe can be accurately determined using the Canny edge detector [28], which returns a binary image with detected edges of thickness 1px. Furthermore, in the studies conducted [29], it was recognized as the most effective among the applied and tested algorithms. In Figure 5, recognized contours are depicted, which have been thickened for better visibility.

As can be observed, the image displays not only the edges of the pipes but also other objects, such as impurities, on the measurement table. These can be discarded by overlaying the previously determined binary mask image of the pipes onto the image with edges.

Notice that the edge detector has also detected light reflections inside the contours of the pipes. We discard them by applying a shadowed binary mask, which we then invert and overlay onto the edge image again. As a result, we obtain an image in which only the contours of the bent pipes remain (Figure 6).

Finally, the edge points in the pixel coordinates are converted to the metric coordinates expressed in the coordinate system of the measurement table $X_{t}Y_{t}Z_{t}$. For this purpose, a dedicated calibration model of the measurement station was utilized. This model was developed by the authors due to the impracticality of using traditional calibration methods, such as Zhang’s method with a 2D pattern, for large-scale setups. The industrial measurement table is over 6 m long, which makes it difficult to fabricate and position a sufficiently large and rigid 2D calibration pattern at various angles and distances from the camera. Therefore, the proposed calibration model enables an unambiguous mapping between the pixel coordinates of the image and the metric coordinates of the measurement table, using a 2D pattern placed flat on the table surface and arranged in various positions on the table plane. The model accounts for perspective effects and camera nonlinearities, including lens distortion, and its parameters can be determined uniquely. A comprehensive description and validation of this calibration model is beyond the scope of this paper.

### 2.2. Determining Pipe Parameters

The pipe is modeled as a composition of cylindrical and toroidal surfaces with a constant cross-section and bend radius in the inspected segment. The method targets dimensional verification of nominal geometry and does not estimate local defects such as cross-sectional ovalization, wrinkling, or gradually varying bend radii.

The determination of the parameters of the bent pipes is divided into three stages: (1) determination of two pairs of parallel lines (projections of straight segments of the pipe); (2) determination of curves that represent the projection of a torus fragment; (3) fitting of the pipe parameters using the analytical model presented in the work [1].

#### 2.2.1. Modification of the Hough Transform

An effective method to reduce errors is to fit a line or curve to the data obtained. For this purpose, the line equations were determined using an algorithm based on the Hough transform [30]. In the case of a line, its position will be determined in polar coordinates.(1)xcosθ+ysinθ=ρ.

The plot of this equation for a single point is a sinusoid. Angle θ∈[0,π] and distance $\rho \in [-\rho_{\max},\rho_{\max}]$, where the value of $\rho_{\max}$ is the distance from the origin of the coordinate system to the farthest point (for an image, this is the length of the diagonal).

By solving Equation (Equation 1) for a chosen image point (x,y), we determine the distance ρ to the line with the inclination angle ${\hat{\theta }}_{1}$ passing through this point. We round the obtained value of ρ to the nearest discrete value ${\hat{\rho }}_{j}$, and then we increment the value of the accumulator cell $A_{h}$ corresponding to the determined pair of parameters ${\hat{\theta }}_{1}$, ${\hat{\rho }}_{j}$ by one. We perform these calculations for all the points on the image. Then, we repeat the scheme for the remaining discrete values of the line inclination angle ${\hat{\theta }}_{i}$.

We can find the parameters of the lines by searching for local maxima, where the value corresponds to the number of points lying on that line. Notice that the projection of a straight segment of a pipe lying on the table consists of a pair of parallel lines, thus having the same inclination angle ${\hat{\theta }}_{i}$ but different distances ${\hat{\rho }}_{i,1}$ and ${\hat{\rho }}_{i,2}$. Such a pair of lines can be found, for example, by solving the following task:(2)$$\max_{{\hat{\theta }}_{i},{\hat{\rho }}_{i,1},{\hat{\rho }}_{i,2}}\left(\Pi_{h}\left({\hat{\theta }}_{i},{\hat{\rho }}_{i,1}\right)+\Pi_{h}\left({\hat{\theta }}_{i},{\hat{\rho }}_{i,2}\right)\right),$$
assuming that the distance between the lines $|{\hat{\rho }}_{i,1}-{\hat{\rho }}_{i,2}|$ is sufficiently large (e.g., $|{\hat{\rho }}_{i,1}-{\hat{\rho }}_{i,2}|>\delta_{\rho }$, where $\delta_{\rho }$ is not greater than the nominal diameter *D* of the examined pipes).

During the search for pairs of lines, points close to the line within a distance η were additionally considered, and the matching index was defined as a function:(3)$$g\left({\hat{\theta }}_{i}\right)=\max_{{\hat{\rho }}_{i,1},{\hat{\rho }}_{i,2}}\left(\sum_{j=-\eta }^{\eta }\Pi_{h}\left({\hat{\theta }}_{i},{\hat{\rho }}_{i,1}+j\right)+\sum_{j=-\eta }^{\eta }\Pi_{h}\left({\hat{\theta }}_{i},{\hat{\rho }}_{i,2}+j\right)\right)$$

The parameter η was derived from the effective pixel size at the working distance to reduce the impact of image noise on line fitting. The separation threshold $\delta_{\rho }$ is set below the minimum nominal pipe diameter in the inspected family to reject spurious parallel pairs.

By solving Equation (Equation 3) for a sample pipe with parameters η=0.1mm and $\delta_{\rho }=10.0\;\mathrm{mm}$, the quality index curve can be obtained as shown in Figure 7. Two maxima are visible for the angles $\theta_{1}$ and $\theta_{2}$. Furthermore, in the algorithm, it was assumed that the difference between the angles $|\theta_{1}-\theta_{2}|$ should be greater than the minimum bending angle $\delta_{\theta }$ occurring in production, which is 0.5°.

The result of the described algorithm is the parameters of two pairs of parallel lines $\left(\theta_{1},\rho_{1,1},\rho_{1,2}\right)$ and $\left(\theta_{2},\rho_{2,1},\rho_{2,2}\right)$, which are visible in the Hough plane and are marked in Figure 8. The two pairs of parallel lines determined have been overlaid on the edges of the pipe in Figure 9.

#### 2.2.2. Determination of the Diameter and Bending Angle of the Pipe

The next task is to determine the centerline for the cylindrical segments of the pipe.

To determine the parameters of the pipe, we first seek the equations of the centerlines of the two cylindrical segments of the pipe. Consider a cylindrical segment of the pipe with an angle of inclination $\theta_{1}$. Consider three planes: the table plane $\pi_{1}$ and two tangent planes to the cylinder passing through the camera point $\pi_{2}$, $\pi_{3}$, and then the perpendicular section to the three described planes passing through the camera point C (Figure 10).

Our goal is to determine the radius of the circle inscribed in the triangle ABC and the coordinate $\rho_{1}$ of the orthogonal projection of the center of the circle on the table plane, using the fact that the coordinates of the points A and B are equal to $\rho_{1,1}$ and $\rho_{1,2}$, respectively. We have the following:(4)$$r_{1}=\frac{\left|\mathrm{AB}\right|z_{c}}{\left|\mathrm{AB}|+|\mathrm{BC}|+|\mathrm{CA}\right|},$$
where $\left|\mathrm{AB}|=\right|\rho_{i,1}-\rho_{i,2}|$, $\left|\mathrm{BC}\right|=\sqrt{z_{c}^{2}+\rho_{1,2}^{2}}$, $\left|\mathrm{CA}\right|=\sqrt{z_{c}^{2}+\rho_{1,1}^{2}}$.

Similarly, we will determine the radius value $r_{2}$. Of course, in reality, the radius of the pipe at both ends is the same. Therefore, we can assume that(5)$$r=\frac{r_{1}+r_{2}}{2}.$$

From Figure 10, we can also deduce the method for determining the coordinate $\rho_{1}$. We have (assuming $r_{1}=r$)(6)$$\rho_{1}=\rho_{1,1}+r\cot\left(\frac{1}{2}∢\mathrm{BAC}\right)$$

Similarly, we will determine the centerline parameter of the second cylindrical segment.

As a result of the calculations, we obtained the parameters of the orthogonal projection of the centerlines on the table plane for both cylindrical segments of the pipe $\left(\theta_{1},\rho_{1}\right)$ and $\left(\theta_{2},\rho_{2}\right)$, as shown in Figure 11.

Of course, on the lines passing through the centerline of the cylindrical segments of the pipe, we can describe the bent pipe in four different orientations (Figure 12). We will make the choice based on the midpoint of the pipe outline $P_{n}$, which is the result of the arithmetic mean of the edge points of the pipe $P_{e}$; let us denote it as follows:(7)$$P_{n}=\left[\begin{array}{c} x_{n}\\ y_{n} \end{array}\right]=\sum_{i=i}^{n_{p}}\frac{P_{e,i}}{n_{p}},$$
where *i* is the ordinal number of the edge point and $n_{p}$ denotes the total number of these points.

Next, for the parameters of the centerlines $\left(\theta_{1},\rho_{1}\right)$ and $\left(\theta_{2},\rho_{2}\right)$, we will adopt new labels: $\left(\theta_{H},\rho_{H}\right)$ for the line with a greater inclination angle and $\left(\theta_{L},\rho_{L}\right)$ for the line with a smaller inclination angle.

Using Equation (Equation 1), we determine the distances $\rho_{n,H}$ and $\rho_{n,L}$ to the lines that pass through the midpoint $P_{n}$ at angles $\theta_{H}$ and $\theta_{L}$.

Based on the values obtained, we can determine the assessment of the bending angle.(8)$$\gamma =\left\{\begin{array}{cc} \theta_{H}-\theta_{L} & \mathrm{for}\;\rho_{n,H}>\rho_{H}\;\wedge \;\rho_{n,L}>\rho_{L}\\ \pi -(\theta_{H}-\theta_{L}) & \mathrm{for}\;\rho_{n,H}>\rho_{H}\;\wedge \;\rho_{n,L}<\rho_{L}\\ \theta_{H}-\theta_{L} & \mathrm{for}\;\rho_{n,H}<\rho_{H}\;\wedge \;\rho_{n,L}<\rho_{L}\\ \pi -(\theta_{H}-\theta_{L}) & \mathrm{for}\;\rho_{n,H}<\rho_{H}\;\wedge \;\rho_{n,L}>\rho_{L} \end{array}\right.,$$
and determine the inclination angle of the orthogonal projection of the centerline of the pipe at the starting point $P_{0}$ relative to the OX axis:(9)$$\beta_{1}=\left\{\begin{array}{cc} \theta_{L}-\pi /2 & \mathrm{for}\;\rho_{n,H}>\rho_{H}\;\wedge \;\rho_{n,L}>\rho_{L}\\ \theta_{H}-\pi /2 & \mathrm{for}\;\rho_{n,H}>\rho_{H}\;\wedge \;\rho_{n,L}<\rho_{L}\\ \theta_{L}+\pi /2 & \mathrm{for}\;\rho_{n,H}<\rho_{H}\;\wedge \;\rho_{n,L}<\rho_{L}\\ \theta_{H}+\pi /2 & \mathrm{for}\;\rho_{n,H}<\rho_{H}\;\wedge \;\rho_{n,L}>\rho_{L} \end{array}\right..$$

#### 2.2.3. Identification of the Bending Radius of the Pipe

To determine the bending radius *R* of the pipe, consider the perspective projection of its bent segment, described by the complex equations of the analytical model: Equation (Equation 12) in Ref. [1]. Note that the center of the torus segment $O_{1}$ lies on the bisector of the adjacent angle γ formed by the lines lying on the centerline of the two cylindrical parts of the pipe (Figure 13), and is also offset from them by a distance *R*. We will determine the position of the point $O_{1}$ based on the real perspective projection of the bent pipe edge. Let us assume that the distance of point $O_{1}$ from the centerline, i.e., the value *R*, belongs to a certain interval $\left[R_{\min},R_{\max}\right]$. By discretizing the interval with a step $\delta_{R}$, we obtain a sequence of values ${\hat{R}}_{i}$.

Whenever the value of ${\hat{R}}_{i}$ changes, it is necessary to recalculate the parameters of the mathematical model. For simplicity, assume that the lengths of the straight sections $l_{1}=0$ and $l_{2}=0$ (their correct values will be determined in subsequent steps). Based on Figure 13, we determine the point.(10)$$P_{1}=P_{s}-d\left[\begin{array}{c} \cos\beta_{1}\\ \sin\beta_{1} \end{array}\right],\;d=\hat{R}\tan\frac{\gamma }{2},$$
where *d* is the offset distance along the centerline and $P_{s}$ is the intersection point of the centerlines of both cylindrical segments of the pipe with parameters $\left(\theta_{1},\rho_{1}\right)$ and $\left(\theta_{2},\rho_{2}\right)$.

Next, define two accumulators $A_{1}$ and $A_{2}$ as vectors with initial values set to zero, where the cell index *i* is associated with the discrete values ${\hat{R}}_{i}$. For each edge point, we calculate the distance from this point to the two theoretical edge lines determined from the analytical model. If this distance is less than $\delta_{R}$, we increase the value of the accumulator $A_{1,i}$ or $A_{2,i}$ accordingly. The value by which we increase the accumulators is 1 when the points coincide, and decreases linearly to zero once the distance $\delta_{R}$ between them is reached.

Finally, we determine the index $i^{*}$, which is the solution to the task.(11)$$\max_{i}=A_{1,i}+A_{2,i},$$
and then we adopt the final assessment of the bending radius of the pipe as(12)$$R={\hat{R}}_{i^{*}}.$$

Figure 14 shows the change in the matching assessment indicator $A_{1,i}+A_{2,i}$ determined for the adopted sequence of ${\hat{R}}_{i}$ values of the investigated bending radius of the pipe. The comparison of the fitted bent arc to the measured edge points of the pipe is presented in Figure 15, where the centerline of the pipe arc, its starting and ending points $P_{1}$ and $P_{2}$, and the bending center of the pipe $O_{1}$ are also marked.

#### 2.2.4. Identification of the Lengths of Pipe Segments

The remaining parameters sought are the lengths of the straight segments of the pipe, denoted $l_{1}$ and $l_{2}$. Let us begin by making a rough estimate of these lengths. To achieve this, we will fit two lines to the edge points at the ends of the pipe with parameters $\left(\theta_{l,1},\rho_{l,1}\right)$ and $\left(\theta_{l,2},\rho_{l,2}\right)$, which (ideally) intersect with the lines passing through the centerline of the cylindrical segments of the pipe at right angles, meaning that their inclinations are(13)$$\theta_{l,1}=\theta_{1}\pm \frac{\pi }{2},\;\theta_{l,2}=\theta_{2}\pm \frac{\pi }{2}.$$

The remaining parameters of the sought lines—the distances ${\hat{\rho }}_{l,1}$ and ${\hat{\rho }}_{l,2}$—are found by searching for local maxima on the Hough plane $\Pi_{h}\left(\theta ,\rho \right)$ as described in Section 2.2.1 and depicted in Figure 16. Additionally, we will consider points lying at a distance η from the line by solving the following task:(14)$$\max_{{\hat{\rho }}_{l,i}}\left(\sum_{j=-\eta }^{\eta }\Pi_{h}\left(\theta_{l,i},{\hat{\rho }}_{l,i}+j\right)\right).$$

Before solving the task, remember to reset the previously identified local maxima and their closest neighborhood on the Hough plane $\Pi_{h}\left(\theta ,\rho \right)$.

In Figure 16, the parameter values for the determined lines that pass through the edge points at the ends of the pipe are marked. These lines are also shown in Figure 17.

Based on them, we will determine the intersection points with the already known lines representing segments of the centerline of the pipe and consider them as approximations of points $P_{0}$, $P_{3}$, representing the positions of the extreme transverse sections of the pipe.

Therefore, the approximate lengths of the straight segments of the pipe are as follows:(15)$${\tilde{l}}_{1}=|{\tilde{P}}_{0}P_{1}|,\;{\tilde{l}}_{2}=\left|P_{2}{\tilde{P}}_{3}\right|.$$

Continuing with the task, we will determine the final estimate of the length of the pipe segments. Let us start by determining the length of the first straight segment of the pipe, denoted as $l_{1}$. To achieve this, we will again apply the mathematical model Equation (Equation 12) in Ref. [1] to determine the projection fragment of the perspective for the first end of the pipe and then to assess its fit to the measured edge points.

Let us assume that the length of the segment $l_{1}$ is sought within a certain discrete interval ${\hat{l}}_{1}=\left[{\hat{l}}_{\min},{\hat{l}}_{\max}\right]$ with a discretization step $\delta_{l}=0.1\;\mathrm{mm}$, where the approximate length of the segment ${\tilde{l}}_{1}$ is one of the elements of the sequence ${\hat{l}}_{1}$.

Furthermore, let us define the accumulator $A_{1}$ as a vector with initial zero values, where its elements are associated with discrete values ${\hat{l}}_{1,i}$.

Each time the value of ${\hat{l}}_{1,i}$ changes, we assume a new position for the point $P_{0}$, which defines the beginning of the centerline in the pipe model. Let us express this as follows: (16)$$P_{0}=P_{1}-\left[\begin{array}{c} \cos\beta_{1}\\ \sin\beta_{1} \end{array}\right]{\hat{l}}_{1,i}.$$

For each edge point located at the beginning of the pipe, let us calculate its distance from the pipe contour segment determined by the mathematical projection model Equation (Equation 12) described in Ref. [1]. If this distance is less than $\delta_{l}$, we increase the corresponding cell of the accumulator $A_{1}$ by a value of the range $(0,1\rangle$, which is inversely proportional to the distance between points in the range $(0,\delta_{l}\rangle$.

After evaluating the fit for all values in the interval ${\hat{l}}_{1}$, we determine the index of the cell $i^{*}$ for which the accumulator $A_{1}$ reaches its maximum value. Finally, let us adopt the measured length of the first segment.(17)$$l_{1}={\hat{l}}_{1,i^{*}}.$$

In Figure 18, the contents of the accumulator $A_{1}$ and the solution sought are presented. The comparison between the fit outline of the beginning of the pipe and the edge points of the pipe is shown in Figure 19.

The following is a note on visualization. In a single-view perspective projection of a thick object, the centerline point $P_{0}$ need not lie on the visible outer contour due to finite thickness and viewing geometry. The observed offset in Figure 19 is consistent with the projection of the pipe end (in red) relative to the detected edge (in blue).

The last parameter sought is the length of the second segment of the pipe, denoted as $l_{2}$. We seek it in a manner analogous to the length of the first segment already presented. However, this time we do not specify the position of the end point of the centerline of the pipe $P_{4}$ because, in the mathematical model, the position of this point is an output value.

The calculated values of the matching indicator for the sequence of values ${\hat{l}}_{2,i}$ are stored in the cells *i* of a new accumulator $A_{2}$. Then, we again search for the index of the cell $i^{*}$ for which the value stored in the accumulator is highest. Finally, we adopt the length of the second segment as follows: (18)$$l_{2}={\hat{l}}_{2,i^{*}},$$
where the index $i^{*}$ is the solution to the optimization problem $\max_{i}\left(A_{2,i}\right)$.

For the discussed example, the content obtained from the accumulator $A_{2}$ is shown in Figure 20. The final fitting segment of the tube along with the last point $P_{3}$ on the centerline of the pipe is shown in Figure 21.

We have determined all the parameters of the pipe with a single bend: the diameter of the pipe D, the bending radius R, the bending angle γ, and the lengths of the cylindrical segments of the pipe $l_{1}$ and $l_{2}$. The theoretical outline of the pipe calculated from the mathematical model was compared with the measured edge points of the pipe, presented in Figure 22.

The final result for the example discussed is presented in Figure 23. The figure shows the undistorted camera image with the pipe, the theoretical perspective projection of the pipe edge calculated using the analytical model (shown in red), and the determined pipe parameters. The parameters shown are the pipe diameter D=62.2mm, the bending angle γ=89.95°, the bending radius R=238.6mm, the lengths of the cylindrical segments of the pipe $l_{1}=99.6\;\mathrm{mm}$ and $l_{2}=295.0\;\mathrm{mm}$, and the total pipe length $l_{m}=769.2\;\mathrm{mm}$.

Similarly, parameters can be determined for pipes with two bends in the same plane.

## 3. Results and Discussion

In this research paper, the authors addressed the issue of controlling the bending parameters of large-diameter pipes. This involves the bending angle and radius and also determining straight sections, which is carried out by algorithms of modeling using methods such as photogrammetry and digital image processing. All proposed algorithms were implemented and used. For this purpose, a single-camera vision measurement system was built and checked by comparing its measurement results with the CMM machine results. The system showed high accuracy and repeatability of the measurements, which is sufficient for heavy industry. In Table 1 are results for the accuracy of the proposed vision system compared to the measurement performed on the Zeiss Prismo Navigator CMM. Notably, the bending radius exhibits a standard deviation of 0.87mm. Compared with the literature review and the fact that only one camera exists in the system, the result is the advantage of the designed system.

The method of determining the shape of the pipe is shown based on the perspective projection and modified Hough transform. The vast majority of the paper describes the algorithms for determining the straight elements of the bent pipe and the correct detection of the pipe ends. It is worth mentioning that this element of the measurement is difficult to estimate due to the smooth transition between the straight section and the bending curve and the contour noise.

The presented work together with the work [1] is a complete and detailed mathematical description of the measurement system with respect to perspective modeling and identification of measurement parameters.

The measurement accuracy achieved with the proposed single-view vision system is lower than that offered by advanced multicamera solutions. However, the algorithms implemented exhibit strong resistance to image noise, which is particularly advantageous in industrial environments. Importantly, for applications in the heavy industry, the level of measurement inaccuracy remains acceptable, as dimensional tolerances depend on the scale of the inspected components. In the case of large-scale bent pipes, allowable deviations typically amount to several millimeters, which aligns with the precision provided by the developed system.

Future research should consider the application of lenses with a higher modulation transfer function (MTF) to improve image fidelity. In addition, developing a method for shadow removal under pipes could further enhance the reliability and accuracy of the inspection process.

## Figures and Tables

**Figure 1 sensors-25-05420-f001:**
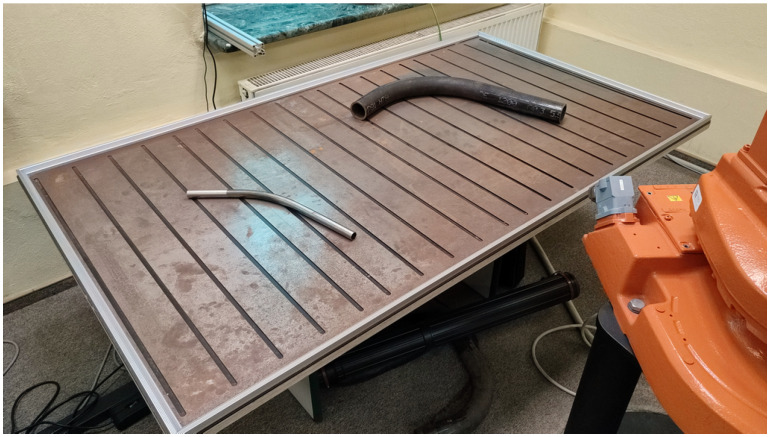
Side view of the laboratory measurement station. The surface of the table is designed to replicate the segmented marking table used in real industrial conditions.

**Figure 2 sensors-25-05420-f002:**
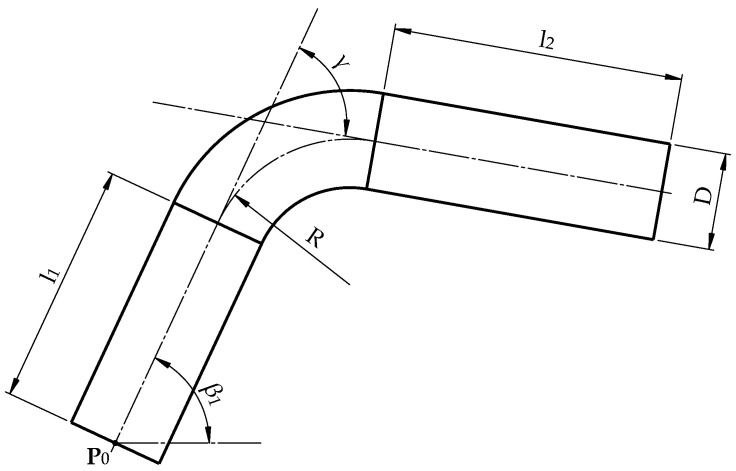
Parameters of the bent pipe—bending angles γ, bending radius R, lengths of straight sections $l_{1}$, $l_{2}$, pipe diameter D=2r, initial point of the centerline $P_{0}$, the slope of the centerline at point $P_{0}$ to axis X: $\beta_{1}$.

**Figure 3 sensors-25-05420-f003:**
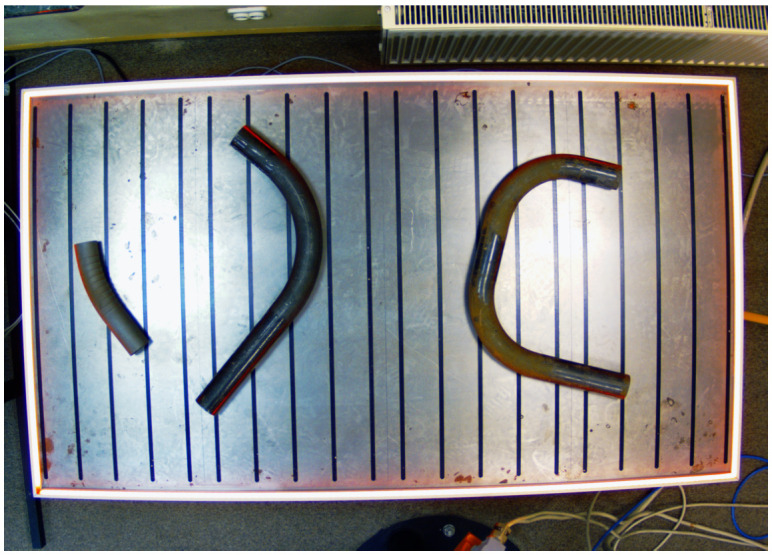
Camera image of the pipe acquired at the measurement station after histogram equalization.

**Figure 4 sensors-25-05420-f004:**
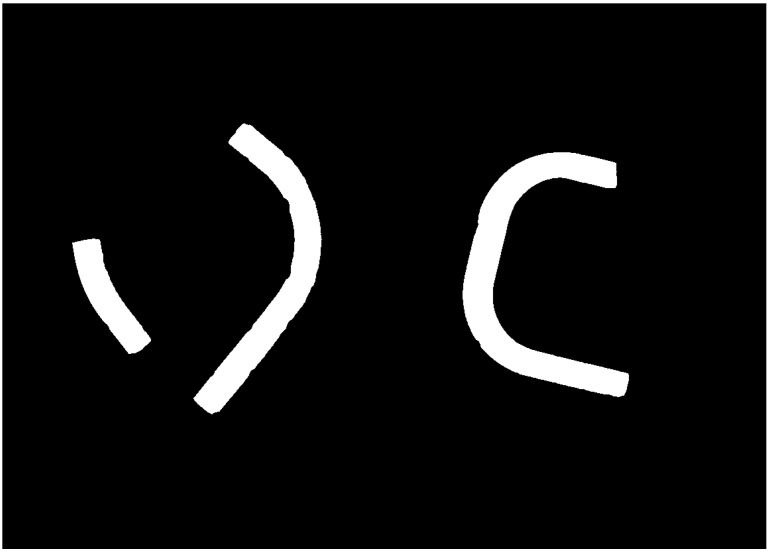
Binary masks of the pipes obtained through image segmentation, including background subtraction, morphological processing, and binarization.

**Figure 5 sensors-25-05420-f005:**
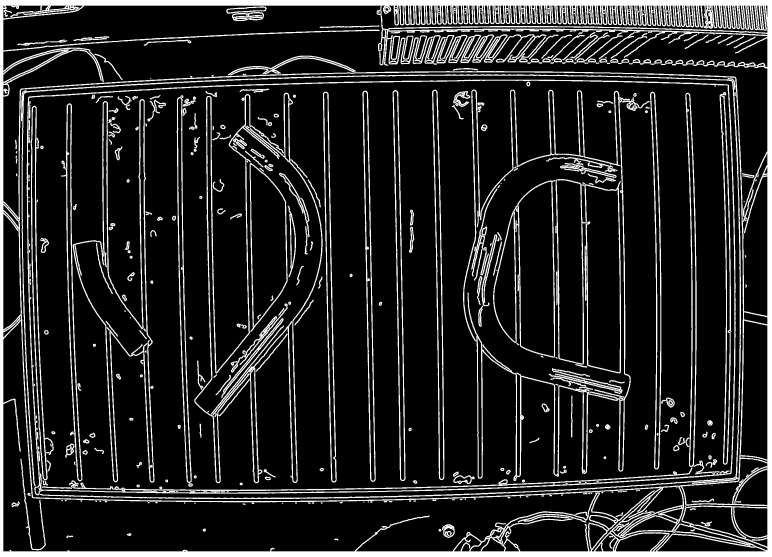
Camera image after edge detector.

**Figure 6 sensors-25-05420-f006:**
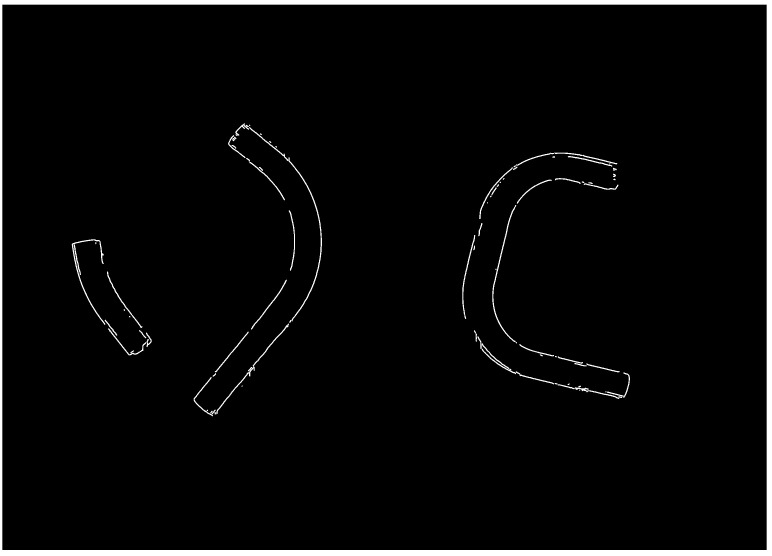
Detected pipe edges in the image, obtained using an edge detector and selected by filtering within the regions of the binary pipe masks.

**Figure 7 sensors-25-05420-f007:**
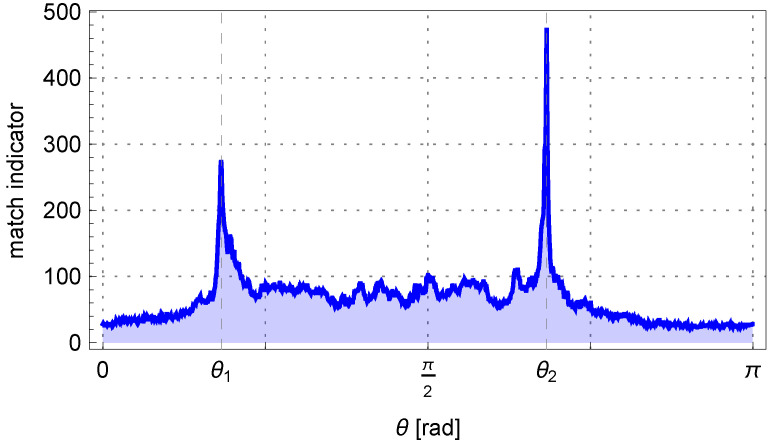
The matching indicator for a pair of parallel lines depends on the angle θ.

**Figure 8 sensors-25-05420-f008:**
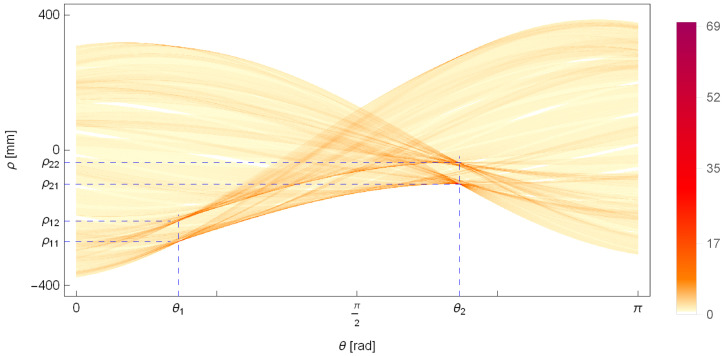
The Hough plane, showing the number of points lying on a line with an inclination angle θ and a distance ρ from the origin of the coordinate system. The detected two pairs of parallel lines with parameters $\left(\theta_{1},\rho_{1,1},\rho_{1,2}\right)$ and $\left(\theta_{2},\rho_{2,1},\rho_{2,2}\right)$ are marked on the plot.

**Figure 9 sensors-25-05420-f009:**
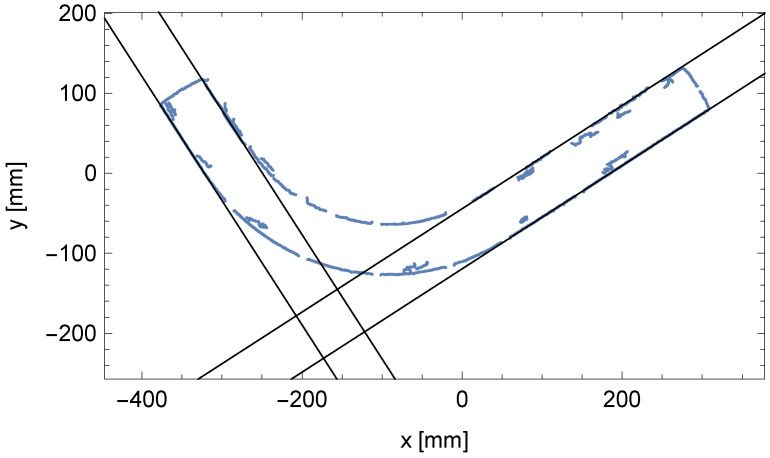
Matched pairs of parallel lines to the edge points of cylindrical segments of the bent pipe.

**Figure 10 sensors-25-05420-f010:**
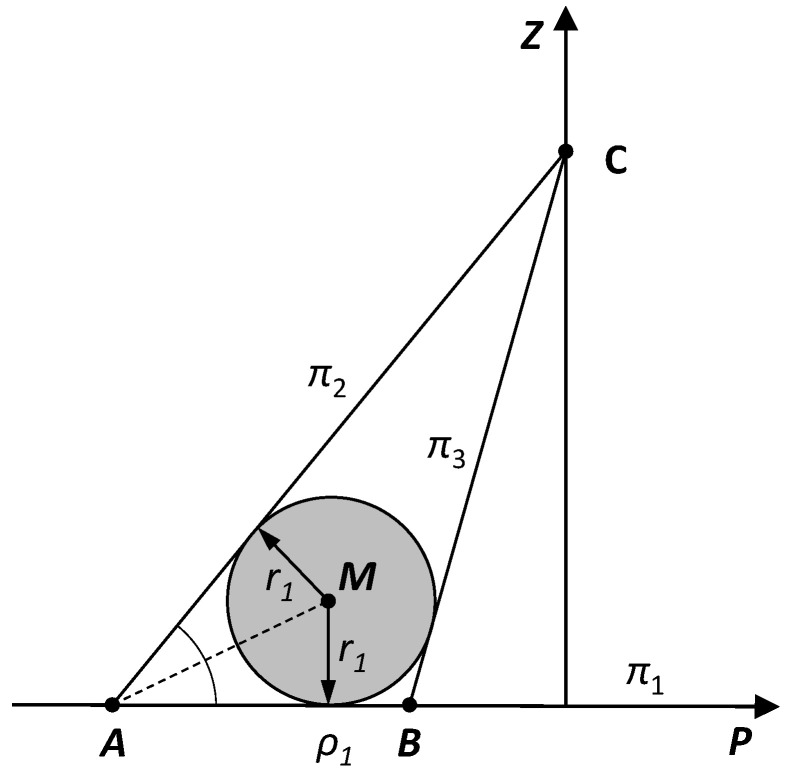
The cross-section of the pipe with visible edge points A and B from camera point C and the point on the centerline M.

**Figure 11 sensors-25-05420-f011:**
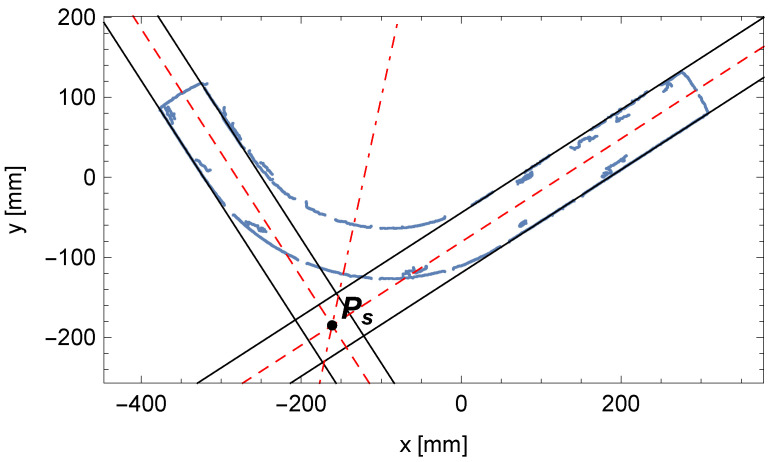
The orthogonal projection of the determined centerlines (dashed lines) for cylindrical segments of the pipe with parameters $\left(\theta_{1},\rho_{1}\right)$, $\left(\theta_{2},\rho_{2}\right)$, and the bisector of the angle between them with their intersection point $P_{s}$.

**Figure 12 sensors-25-05420-f012:**
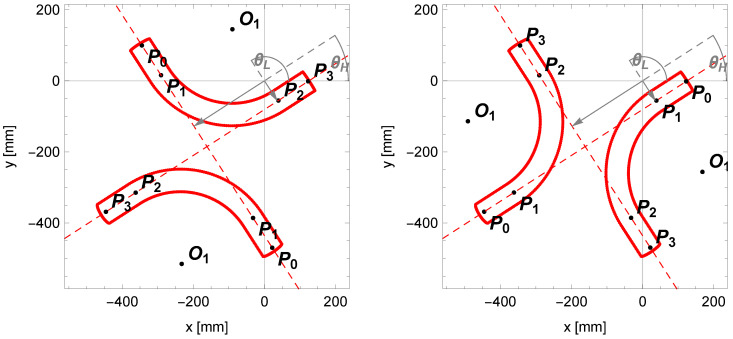
Four pipe configurations determined based on the centerlines with parameters $\left(\theta_{H},\rho_{H}\right)$ and $\left(\theta_{L},\rho_{L}\right)$.

**Figure 13 sensors-25-05420-f013:**
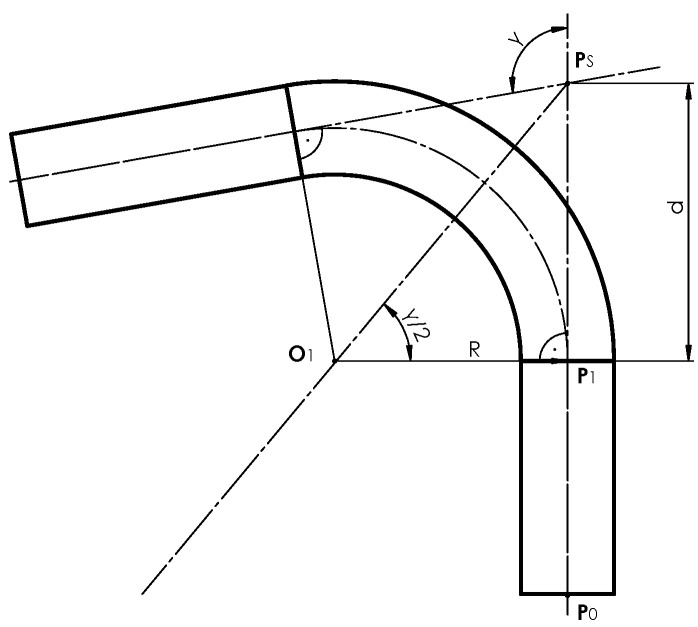
The cross-section of the pipe with the centerline and the bending center point $O_{1}$ lying on the bisector of the angle adjacent to the bending angle of the pipe γ.

**Figure 14 sensors-25-05420-f014:**
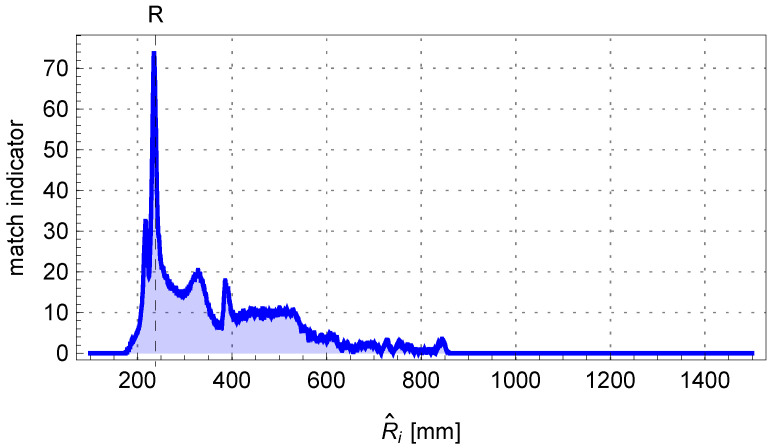
The matching indicator of the edge points of the pipe to the theoretical outline of the bent arc determined from the perspective model.

**Figure 15 sensors-25-05420-f015:**
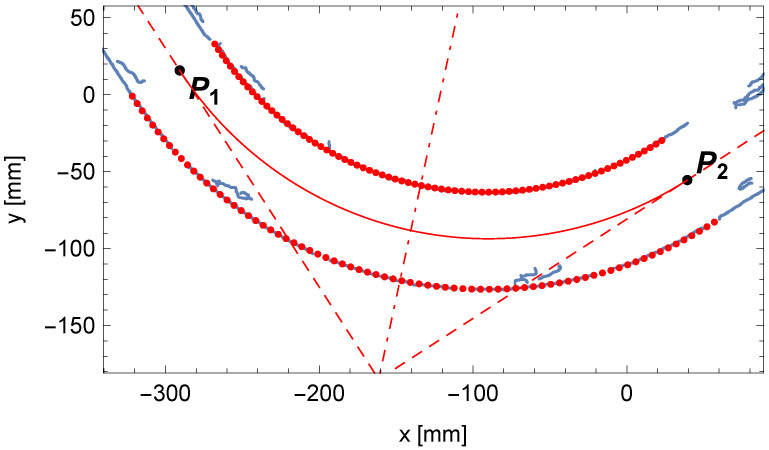
Comparison between the determined outline of the perspective projection of the bent arc (in red) and the edge points of the pipe (in blue).

**Figure 16 sensors-25-05420-f016:**
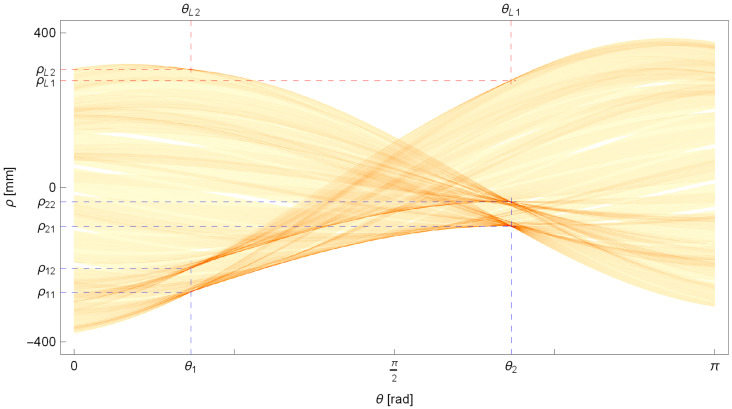
The Hough plane with the determined parameters of the lines $\left(\theta_{l,1},\rho_{l,1}\right)$ and $\left(\theta_{l,2},\rho_{l,2}\right)$ passing through the edge points at the ends of the pipe (in red), and the previously determined parameters of the lines passing through the edge points of the cylindrical sections of the bent pipe (in blue).

**Figure 17 sensors-25-05420-f017:**
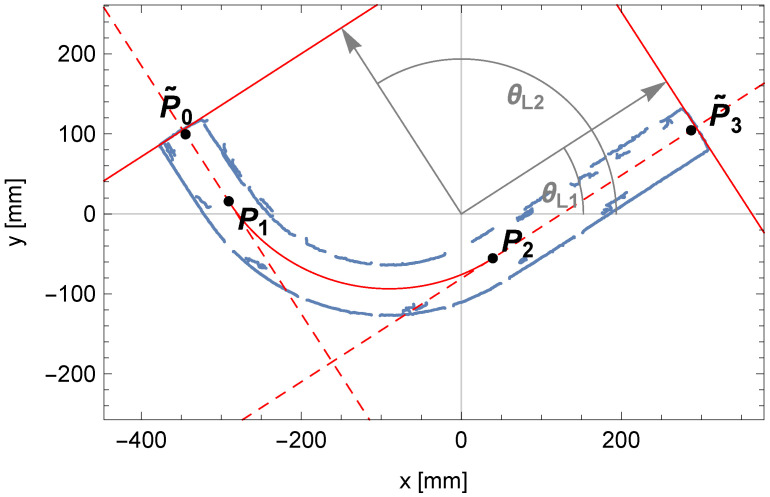
Approximate positions of the pipe endpoints, ${\tilde{P}}_{0}$ and ${\tilde{P}}_{3}$, determined based on lines (in red) passing through the edge points (in blue) at the pipe ends. Both lines are defined by direction vectors (in gray) with leading angles $\theta_{l,1}$ and $\theta_{l,2}$.

**Figure 18 sensors-25-05420-f018:**
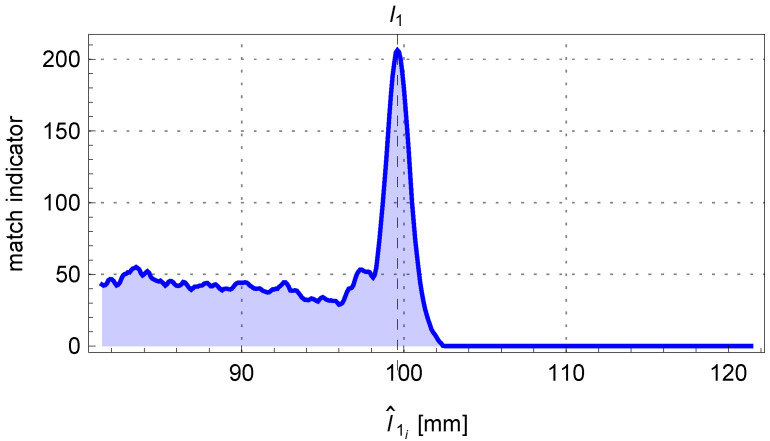
The trend of the matching indicator calculated from the mathematical model’s outline of the beginning of the pipe to the edge points of the pipe as the length $l_{1}$ changes.

**Figure 19 sensors-25-05420-f019:**
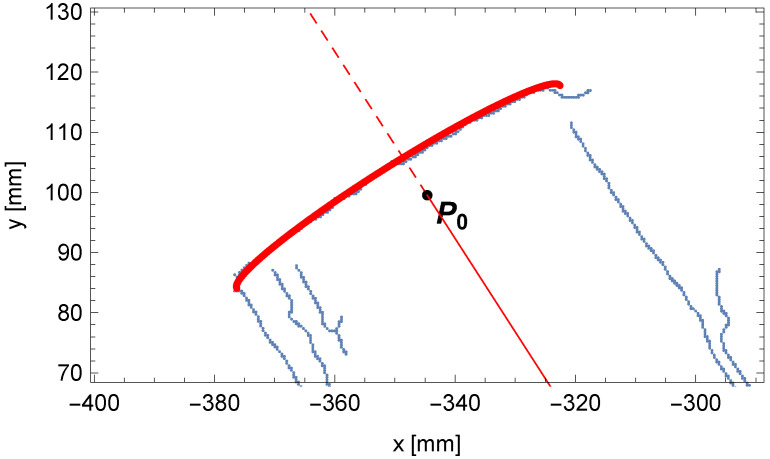
Comparison between the outline of the perspective projection for the beginning of the pipe (in red) and the edge points of the pipe (in blue).

**Figure 20 sensors-25-05420-f020:**
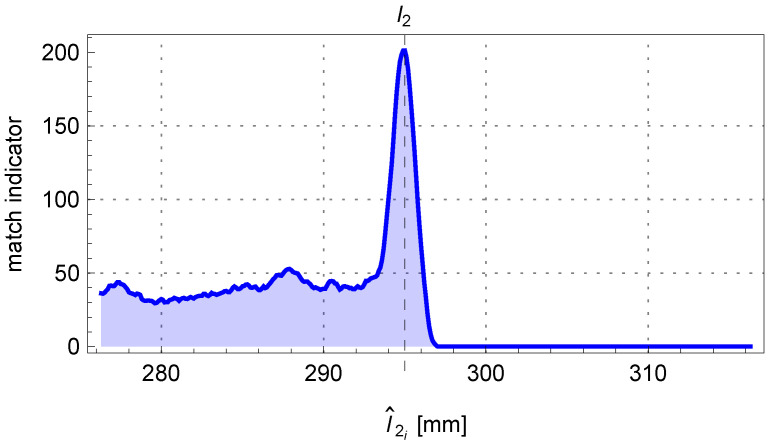
The trend of the matching indicator of the perspective projection outline fragment of the end of the pipe to the edge points of the pipe.

**Figure 21 sensors-25-05420-f021:**
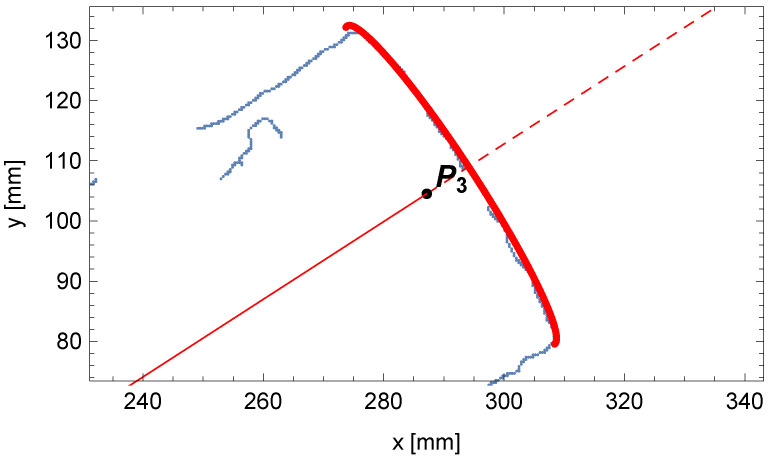
Comparison between the outline of the perspective projection for the end of the pipe (in red) and the edge points of the pipe (in blue).

**Figure 22 sensors-25-05420-f022:**
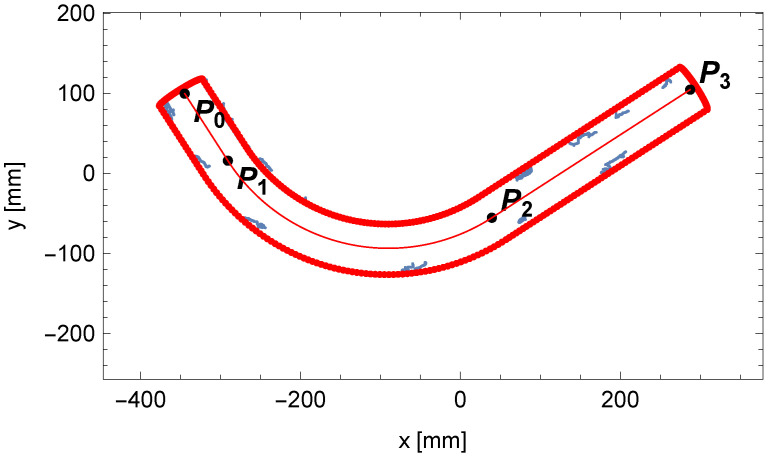
Comparison between the edge points of the pipe (in blue) and the calculated perspective projection of the pipe from the analytical model (in red).

**Figure 23 sensors-25-05420-f023:**
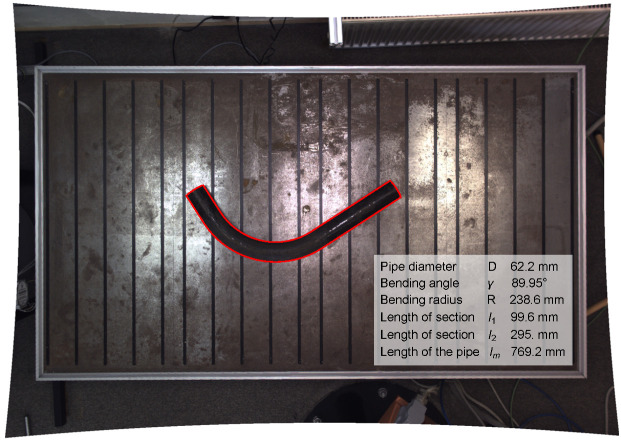
Undistorted camera image showing the pipe, the perspective projection of the pipe edge calculated using the analytical model (in red), and the determined pipe parameters.

**Table 1 sensors-25-05420-t001:** The results of the measuring system accuracy series of measurements of a reference pipe compared to the reference values measured by a Zeiss Prismo Navigator CMM.

		CMM Value	Average Value	Standard Deviation	Relative std. dev.
Pipe diameter	D[mm]	60.8	61.1	0.46	0.76%
Bending angle	γ[°]	89.9	89.8	0.29	0.32%
Bending radius	R[mm]	238.6	238.4	0.87	0.36%
Length of section 1	$l_{1}\;\left[\mathrm{mm}\right]$	100.9	101.7	1.79	1.77%
Length of section 2	$l_{2}\;\left[\mathrm{mm}\right]$	296.4	295.5	2.21	0.75%
Length of the pipe	$l_{m}\;\left[\mathrm{mm}\right]$	771.5	770.8	1.82	0.24%

## Data Availability

The raw data supporting the conclusions of this article will be made available by the authors on request.

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
