# Peer review of "Geometric Parameter Identification of Large Bent Pipes Using a Single-View Vision System"

_sensors, 2025, doi:10.3390/s25175420_

Round 1

Reviewer 1 Report

Comments and Suggestions for Authors

This paper presents a single-camera vision system for measuring geometric parameters of large bent pipes using modified Hough transformation and perspective projection modeling. While addressing a practical industrial need, the work suffers from limited technical novelty, insufficient experimental validation, and methodological concerns that significantly constrain its contribution to the machine vision field.

The claimed technical contributions are primarily incremental adaptations of established computer vision techniques rather than fundamental innovations. The modified Hough transformation in Section 2.2.1 represents only a minor variation for detecting parallel line pairs, lacking theoretical depth. The geometric modeling approach oversimplifies real pipe behavior by treating bent pipes as perfect cylindrical and toroidal surfaces, ignoring common industrial deformations such as cross-sectional ovalization and variable bending radii. The perspective projection model shows visible systematic errors, as evidenced in Figure 17 where the determined starting point Pâ‚€ misaligns with the calculated outline, indicating unresolved distortion correction issues.

The experimental framework presents critical limitations that undermine result reliability. Validation relies on only a few pipe samples from a single manufacturer, insufficient for demonstrating robustness across diverse industrial conditions. The accuracy comparison with CMM measurements in Table 1, while showing reasonable precision, lacks essential error analysis including sensitivity to lighting variations, viewing angles, and surface conditions. Key algorithmic parameters (η = 0.1mm, δρ = 10.0mm, δθ = 0.5°) appear arbitrarily chosen without systematic justification or sensitivity analysis. The accumulator-based optimization for radius determination may suffer from discretization errors and local optima, particularly with complex geometries.

The paper lacks computational complexity analysis and processing time benchmarks essential for industrial deployment claims. Shadow and reflection handling mechanisms are mentioned but not quantitatively analyzed, despite being critical for reliable industrial performance. Presentation quality is compromised by poor figure contrast (Figures 4-5), inconsistent mathematical notation, and grammatical errors throughout. The literature review inadequately addresses single-view approaches and recent deep learning advances in pipe inspection, limiting proper positioning within current research.

The work provides no systematic comparison with existing single-view methods or quantitative trade-off analysis between system complexity and measurement accuracy. Real-time performance claims lack supporting evidence, and robustness to common industrial challenges such as contamination and partial occlusion remains unexplored. The geometric assumptions ignore manufacturing tolerances and mechanical deformation realities, potentially compromising accuracy in practical applications.

While this research addresses a relevant industrial problem, the limited technical novelty, narrow experimental validation, and methodological shortcomings significantly diminish its scientific contribution. The work represents an incremental engineering application rather than a fundamental advancement in vision-based measurement. Substantial improvements in experimental scope, technical depth, error analysis, and presentation quality are required before the work meets publication standards for a peer-reviewed materials technology journal.

Author Response

Thank you very much for taking the time to review our manuscript. Please find our point‑by‑point responses below, with the corresponding revisions and corrections highlighted (track changes) in the re‑submitted files. We appreciate the reviewers’ constructive feedback and have addressed each comment carefully.

Comments 1:

This paper presents a single-camera vision system for measuring geometric parameters of large bent pipes using modified Hough transformation and perspective projection modeling. While addressing a practical industrial need, the work suffers from limited technical novelty, insufficient experimental validation, and methodological concerns that significantly constrain its contribution to the machine vision field.

Response 1:

Thank you for the detailed review and the time dedicated to analyzing our work. We agree that the article is application-oriented and focuses on solving a specific industrial problem, which, in our opinion, constitutes its main value. The aim of the research was to develop a practical tool that can be implemented in industrial conditions, rather than to develop entirely new theories in the field of computer vision.

Comments 2:

The claimed technical contributions are primarily incremental adaptations of established computer vision techniques rather than fundamental innovations. The modified Hough transformation in Section 2.2.1 represents only a minor variation for detecting parallel line pairs, lacking theoretical depth. The geometric modeling approach oversimplifies real pipe behavior by treating bent pipes as perfect cylindrical and toroidal surfaces, ignoring common industrial deformations such as cross-sectional ovalization and variable bending radii. The perspective projection model shows visible systematic errors, as evidenced in Figure 17 where the determined starting point Pâ‚€ misaligns with the calculated outline, indicating unresolved distortion correction issues.

Response 2:

Thank you for the critical remarks. We agree that the proposed method builds on established computer vision techniques; however, its originality lies in the purposeful integration of these algorithms with mathematical modeling and the use of prior information about the nominal shape of the measured object. This enabled a practical solution for estimating the geometric parameters of a bent pipe from a single image, which—to the best of our knowledge—has not been reported in the literature.

Regarding the geometric model simplifications, we emphasize that the system is designed for dimensional verification of nominal cylindrical/toroidal geometry and is not intended to detect defects such as ovalization or local variations in bend radius. These limitations are clarified in the revised manuscript (lines 25-27, 217-220).

We do not agree with the allegation of systematic errors in the perspective projection model. The position of the initial point P0 shown in Fig. 17 (now Fig. 18) has been clarified in the revised manuscript (lines 336-339).

Comments 3:

The experimental framework presents critical limitations that undermine result reliability. Validation relies on only a few pipe samples from a single manufacturer, insufficient for demonstrating robustness across diverse industrial conditions. The accuracy comparison with CMM measurements in Table 1, while showing reasonable precision, lacks essential error analysis including sensitivity to lighting variations, viewing angles, and surface conditions. Key algorithmic parameters (η = 0.1mm, δρ = 10.0mm, δθ = 0.5°) appear arbitrarily chosen without systematic justification or sensitivity analysis. The accumulator-based optimization for radius determination may suffer from discretization errors and local optima, particularly with complex geometries.

Response 3:

Thank you for these detailed remarks. Our study was conducted in response to the needs of a local industrial partner, and the algorithmic parameters were selected to match the nominal dimensional range of the inspected pipes. Specifically, δρ was linked to the minimum nominal pipe diameter to suppress spurious parallel-line pairings; δθ reflected the smallest bending angles relevant to production; and η (the edge–line proximity tolerance) was derived from the effective pixel size at the working distance to reduce the impact of image noise on line fitting (added in lines 217–220). The bending-radius estimation does not rely on single points; instead, the accumulator aggregates evidence across many edge points against the theoretical perspective projection of a toroidal segment, which limits the influence of background misdetections and local artifacts. Using this approach, repeated measurements on 20 distinct pipe samples did not exhibit difficulties in reliably determining the bending radius. We agree that broader validation across more diverse samples and illumination conditions would further strengthen generality.

Comments 4:

The paper lacks computational complexity analysis and processing time benchmarks essential for industrial deployment claims. Shadow and reflection handling mechanisms are mentioned but not quantitatively analyzed, despite being critical for reliable industrial performance. Presentation quality is compromised by poor figure contrast (Figures 4-5), inconsistent mathematical notation, and grammatical errors throughout. The literature review inadequately addresses single-view approaches and recent deep learning advances in pipe inspection, limiting proper positioning within current research.

Response 4:

Thank you for the comments. The system is intended for heavy‑industry applications, where unit or small‑batch production and gantry‑crane handling make strict real‑time processing non‑critical. The algorithms were developed and tested in Mathematica (line 104); in the current, non‑optimized form, a single measurement takes approximately 4 minutes, including the generation, visualization, and saving of extensive diagnostic data and images. For industrial deployment, we plan a C++ re‑implementation to substantially reduce processing time; as a prototype, this implementation was designed to empirically confirm sufficient accuracy with a single camera rather than to provide full complexity/benchmark analysis. Figures 4–5 (now 5–6) are black‑and‑white binary images with detected edge lines; their contrast cannot be increased. In addition, we revised and clarified figure captions for improved readability and consistency (now Figures 3, 4, 6, and 22). Edge points have been thickened for readability, and the embedded image quality allows at least 10× magnification. We also expanded Related Works to cover single‑view approaches and recent deep‑learning literature (lines 91–101) and additional background (42-52). Finally, we revised and standardized all mathematical notation and symbol definitions to ensure consistency across sections, figures, and equations.

Comments 5:

The work provides no systematic comparison with existing single-view methods or quantitative trade-off analysis between system complexity and measurement accuracy. Real-time performance claims lack supporting evidence, and robustness to common industrial challenges such as contamination and partial occlusion remains unexplored. The geometric assumptions ignore manufacturing tolerances and mechanical deformation realities, potentially compromising accuracy in practical applications.

Response 5:

Thank you for these remarks. We clarified the scope by explicitly positioning our contribution within single‑view metrology and recent deep‑learning inspection work (lines 91–101). We also note that the manuscript does not claim real‑time operation; the current ~4‑minute runtime reflects a prototype that generates, visualizes, and saves extensive diagnostic data and images. A full benchmark/complexity analysis and a broader robustness study are planned for the engineering implementation. For completeness, we note that the segmentation step may occasionally retain pixels originating from contamination rather than true pipe edges. In practice, the modified Hough transform we employ for line fitting makes the pipeline resilient to such image noise. The current prototype can also handle measuring multiple pipes placed on the table surface, provided they do not overlap. Avoiding overlap is a practical requirement in this application: overlapping large, heavy bent arcs could induce stresses and elastic deformations, potentially biasing measurements and risking damage to the workpieces. Regarding manufacturing tolerances, our present model assumes nominal geometry (lines 187–190). In addition, we introduced an explicit note in the calibration paragraph indicating that a comprehensive description and validation of the calibration model will be presented in a separate publication (lines 175–185).

Comments 6:

While this research addresses a relevant industrial problem, the limited technical novelty, narrow experimental validation, and methodological shortcomings significantly diminish its scientific contribution. The work represents an incremental engineering application rather than a fundamental advancement in vision-based measurement. Substantial improvements in experimental scope, technical depth, error analysis, and presentation quality are required before the work meets publication standards for a peer-reviewed materials technology journal.

Response 6:

Thank you for the assessment. We respectfully submit that, while our contribution is indeed application‑oriented, it addresses a real industrial gap with a single‑camera, single‑image pipeline that delivers explicit geometric parameters (diameter, bend angle, bend radius, straight‑section lengths) for large‑scale bent pipes used in heavy industry and is validated against a CMM. We clarified the methodological scope (nominal cylindrical/toroidal geometry; no defect detection), and provided end‑to‑end algorithms for segmentation, line fitting (modified Hough), and parameter identification. The Results and Discussion include quantitative accuracy metrics and a direct comparison to CMM values, positioning the achieved precision as acceptable for heavy‑industry tolerances. We also expanded the Related Works to situate the paper within single‑view metrology and recent deep‑learning inspection research, and we improved presentation quality (figures and captions). While we acknowledge that the work is not a fundamental theoretical advance, we believe it offers a practical, reproducible solution with clear industrial relevance—consistent with the scope of the journal Sensors, where the manuscript is under consideration. Further broadening of experimental scope, benchmarking, and extended error/robustness analysis are planned for the subsequent engineering implementation.

Reviewer 2 Report

Comments and Suggestions for Authors

The paper is overall well written and worth publishing. However, some points should be considered for improvement:

The introduction should also look at other optical techniques such as patterned light approaches (fringe projection or the like) and why they are not suitable or compatible in view of the authors 

Figure 2 is central for the understanding, however, explanation is scarce: What does one really see? What are the bent lines and what purpsoe do they hev? why are they only in y-direction? Enhance explanation and Figure caption (see also Fig. 21).

Line 144 speaks of a "dedicated calibration model", however, no hint or reference is given. Please elaborate.

Eq.8 and 9 have "dla", wouldnt it be better to replace it by "for" or "for all"

Figure 21 says "The determined parameters of the pipe along with the theoretical, distorted outline of its perspective projection edge presented in the camera image without distortions" This is nowhere founf in the peicture: There are no paremters given, the is no mention ahwt the red line is all about or what the backgound is. Please explain in text and fiure caption: 1. What is to be ssen and 2. what does it mean?

Author Response

Thank you for the thorough and constructive review. We have addressed each comment point by point and updated the manuscript accordingly, with all changes clearly indicated in the resubmitted files. We appreciate your feedback, which helped improve the paper.

Comments 1:

The introduction should also look at other optical techniques such as patterned light approaches (fringe projection or the like) and why they are not suitable or compatible in view of the authors.

Response 1:

Thank you for your suggestion. We have expanded the introduction (lines 42–52) to discuss patterned light approaches and their limitations for our application.

Comments 2:

Figure 2 is central for the understanding, however, explanation is scarce: What does one really see? What are the bent lines and what purpsoe do they hev? why are they only in y-direction? Enhance explanation and Figure caption (see also Fig. 21).

Response 2:

Thank you for this valuable comment. We agree with the reviewer’s suggestion. Accordingly, we have added a detailed explanation (lines 108–113) and included an additional figure with a descriptive caption to further illustrate the setup. The captions for Figures 3–6 and 22 (previously Figures 2–5 and 21) have been revised to provide clearer and more informative descriptions.

Comments 3:

Line 144 speaks of a "dedicated calibration model", however, no hint or reference is given. Please elaborate.

Response 3:

Thank you for your comment. We agree that further clarification is necessary. Additional details have been added (lines 175–185) of the revised manuscript. A comprehensive description and validation of this dedicated calibration model will be presented in a separate publication currently in preparation.

Comments 4:

Eq.8 and 9 have "dla", wouldnt it be better to replace it by "for" or "for all"

Response 4:

We agree with this comment. We have replaced “dla” with “for” in Equations 8 and 9 in the revised manuscript.

Comments 5:

Figure 21 says "The determined parameters of the pipe along with the theoretical, distorted outline of its perspective projection edge presented in the camera image without distortions" This is nowhere founf in the peicture: There are no paremters given, the is no mention ahwt the red line is all about or what the backgound is. Please explain in text and fiure caption: 1. What is to be ssen and 2. what does it mean?

Response 5:

Thank you for this valuable comment. We agree with the remark. We have revised Figure 21 (now Figure 22), which now includes the determined pipe parameters. We also updated the figure caption to clearly explain what is shown in the image, including the meaning of the red line and the background. In addition, we added a clarifying explanation in the main text (lines 357–362).

Round 2

Reviewer 1 Report

Comments and Suggestions for Authors

The issues in the comment have been solved